# Modular Quantization-Aware Training for 6D Object Pose Estimation

**Saqib Javed**[1][†] **, Chengkun Li**[1]**, Andrew Price**[1]**, Yinlin Hu**[2]**, Mathieu Salzmann**[1]
[1] *CVLab, EPFL*  [2] *Magic Leap*

**Reviewed on OpenReview:** `https://openreview.net/forum?id=lIy0TEUou7`

## Abstract

Edge applications, such as collaborative robotics and spacecraft rendezvous, demand efficient 6D object pose estimation on resource-constrained embedded platforms. Existing 6D object pose estimation networks are often too large for such deployments, necessitating compression while maintaining reliable performance. To address this challenge, we introduce Modular Quantization-Aware Training (MQAT), an adaptive and mixed-precision quantization-aware training strategy that exploits the modular structure of modern 6D object pose estimation architectures. MQAT guides a systematic gradated modular quantization sequence and determines module-specific bit precisions, leading to quantized models that outperform those produced by state-of-the-art uniform and mixed-precision quantization techniques. Our experiments showcase the generality of MQAT across datasets, architectures, and quantization algorithms. Additionally, we observe that MQAT quantized models can achieve an accuracy boost ($> 7\%$ ADI-0.1d) over the baseline full-precision network while reducing model size by a factor of $4\times$ or more. `https://saqibjaved1.github.io/MQAT_`

## 1 Introduction

6D pose estimation is the process of estimating the 6 degrees of freedom (attitude and position) of a rigid body and has emerged as a crucial component in numerous situations, particularly in robotics applications such as automated manufacturing (Pérez et al., 2016), vision-based control (Singh et al., 2022), collaborative robotics (Vicentini, 2020) and spacecraft rendezvous (Song et al., 2022). However, such applications typically must run on embedded platforms with limited hardware resources.

These resource constraints often disqualify current state-of-the-art methods, such as ZebraPose (Su et al., 2022), SO-Pose (Di et al., 2021), and GDR-Net (Wang et al., 2021), which employ a two stage approach (detection followed by pose estimation) and thus entail a large memory footprint. By contrast, single-stage methods (Thalhammer et al., 2021; Peng et al., 2019; Song et al., 2020; Hu et al., 2021b; Wang et al., 2022; Chen et al., 2019; Rad & Lepetit, 2017; Hodaň et al., 2020) offer a more pragmatic alternative, yielding models with a good accuracy-footprint tradeoff. Nevertheless, they remain too large for deployment on edge devices.

To address this challenge, CA-SpaceNet (Wang et al., 2022) applies a *uniform* quantization approach to reduce the network memory footprint at the expense of a large accuracy loss; all network layers are quantized to the same bit width, except for the first and last layer.

In principle, *mixed-precision* quantization methods (Cai & Vasconcelos, 2020; Dong et al., 2020; Tang et al., 2022) could demonstrate similar compression with better performance, but they tend to require significant effort and GPU hours to determine the optimal bit precision for each layer. Furthermore, neither mixed-precision nor uniform quantization methods consider the importance of the order in which the network weights or layers are quantized, as searching for the optimal order is combinatorial in the number of, e.g., network layers.

---

[†]Corresponding author: `saqib.javed@epfl.ch`

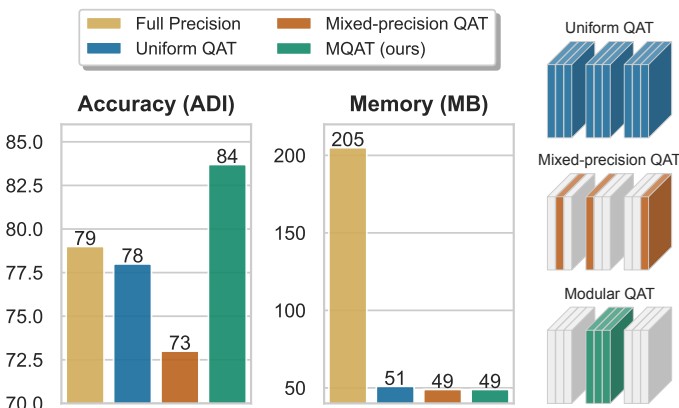

Figure 1: **Summary of this Work.** In contrast to uniform and mixed-precision quantization, MQAT accounts for the modularity of typical 6D object pose estimation frameworks. Uniform QAT quantizes an entire network simultaneously and uniformly; in mixed-precision QAT, layers are quantized to varying bit precisions regardless of their position; in contrast, MQAT applies quantization to network modules in a proposed order, with each module assigned the optimal bit precision. MQAT not only reduces the memory footprint of the network but can result in an accuracy boost that neither uniform nor mixed-precision quantization have demonstrated. The results shown in the figure represent a comparison of different quantization methods applied to the WDR network, evaluated on the SwissCube dataset.

In this work, we depart from such conventional Quantization-Aware Training (QAT) approaches and leverage the inherent modular structure of modern 6D object pose estimation architectures, which typically encompass components such as a backbone, a feature aggregation module, and prediction heads. Specifically, we introduce a Modular Quantization-Aware Training (MQAT) paradigm that relies on a gradated quantization strategy, where the modules are quantized and fine-tuned in a mixed-precision way, following a recommended sequential order (i.e., quantization flow) based on their sensitivity to quantization. As the number of modules in an architecture is much lower than that of layers, it is possible to confirm a quantization flow is optimal.

Our experiments evidence that MQAT is particularly well-suited to 6D object pose estimation, consistently outperforming the state-of-the-art quantization techniques in terms of accuracy for a given memory consumption budget, as shown in fig. 1. We demonstrate the generality of our approach by applying it to different single-stage architectures, WDR (Hu et al., 2021b), CA-SpaceNet (Wang et al., 2022); different datasets, SwissCube (Hu et al., 2021b), Linemod (Hinterstoisser et al., 2012), and Occlusion Linemod (Brachmann et al., 2014); and using different quantization strategies, INQ (Zhou et al., 2017) and LSQ (Esser et al., 2020), within our modular strategy. We further show that our method also applies to two-stage networks, such as ZebraPose (Su et al., 2022), on which it outperforms both uniform and mixed-precision quantization. Furthermore, we extend MQAT to the task of object detection, further validating its efficacy in the computer vision domain. The results of applying MQAT to object detection are presented in appendix A.2, demonstrating its potential for broader applicability.

To summarize, our main contributions are as follows:

- For neural network compression, we are the first to identify and exploit the current trend of modular architectures. We propose Modular Quantization-Aware Training (MQAT), a novel mixed-precision Quantization-Aware Training (QAT) algorithm.

- We further demonstrate that to achieve the best results, the sequence of quantizing modules is significant and thus propose a recommended quantization order. We show that this order is optimal for the quantization of a 3 module 6D pose estimation network (Hu et al., 2021b).

- With MQAT, we show substantial accuracy gains over competitive QAT methods. Furthermore, unlike other QATs, MQAT significantly surpasses full-precision accuracy in the specific case of single-stage 6D pose estimation networks (Hu et al., 2021b; Wang et al., 2022).

- We validate MQAT across different datasets, neural network architectures, underlying quantization algorithms, and tasks, showcasing its adaptability and effectiveness in different settings.

## 2    Related Work

In this section, we survey recent advances in RGB-based 6D object pose estimation, outlining key architectures and their contributions to the field. We then explore the developments in quantization-aware training (QAT), particularly in relation to modular neural network designs. This review sets the groundwork for our proposed Modular Quantization-Aware Training framework, which is inspired by these advances and addresses their limitations.

### 2.1    6D Object Pose Estimation

**Single-Stage Direct Estimation.**    PoseCNN (Xiang et al., 2018) was one of the first methods to estimate 6D object pose using a deep neural network. SSD6D (Liu et al., 2016; Kehl et al., 2017) and URSONet (Proença & Gao, 2020) used a discretization of the rotation space to form a classification problem instead of a regression one.

**Single-Stage with PnP.**    In general, a better performing strategy for convolutional architectures consists of training a network to predict 2D-to-3D correspondences instead of the pose. The pose is then obtained via a RANdom SAmple Consensus (RANSAC) / Perspective-n-Point (PnP) 2D–to–3D correspondence fitting process. These methods typically employ a backbone, a feature aggregation module, and one or multiple heads.(Rad & Lepetit, 2017; Tekin et al., 2018) estimate these correspondences in a single global fashion, whereas (Peng et al., 2019; Jafari et al., 2018; Hu et al., 2019; Zakharov et al., 2019; Markus Oberweger, 2018; Chen et al., 2019) aggregate multiple local predictions to improve robustness. To improve performance in the presence of large depth variations, a number of works (Thalhammer et al., 2021; Hu et al., 2021b; Wang et al., 2022) use an FPN (Lin et al., 2016) to exploit features at different scales.

**Multi-Stage**    The current state-of-the-art pose estimation frameworks incorporate a pipeline of networks that perform different tasks. In the first stage network, the target is localized and a Region of Interest (RoI) is cropped and forwarded to the second stage network. This isolates the position estimation task from the orientation estimation one and further provides the orientation estimation network with an RoI containing only object features. The second stage orientation estimation network can then more easily fit to the target object.

Multi-stage with PnP approaches have generated good results (Su et al., 2022; Di et al., 2021; Wang et al., 2021; Li et al., 2019b; Labbé et al., 2020), but some recent works avoid traditional PnP and RANSAC methods. PViT-6D (Stapf et al., 2023) uses an off-the-shelf object detector to crop the target and then reframes pose estimation as a direct regression task, using Vision Transformers to enhance accuracy. CheckerPose (Lian & Ling, 2023) improves robustness by progressively refining dense 3D keypoint correspondences with a graph neural network. MRC-Net (Li et al., 2024) introduces a two-stage process, combining pose classification and rendering with a multi-scale residual correlation (MRC) layer to capture fine-grained pose details. GigaPose (Nguyen et al., 2024) achieves fast and robust pose estimation through CAD-based template matching and patch correspondences. FoundationPose (Wen et al., 2024) unifies model-based and model-free setups using neural implicit representations and contrastive learning, achieving strong generalization without fine-tuning.

Multi-stage frameworks tend to yield more accurate results. However, they also have much larger memory footprints as they may include one object classifier network; one object position/RoI network; and $N$ object pose networks. For hardware-restricted scenarios, a multi-stage framework may thus not be practical. Even for single-stage networks, additional compression is required (Blalock et al., 2020).

### 2.2    Quantization-Aware Training

Neural network quantization reduces the precision of parameters such as weights and activations. Existing techniques fall into two broad categories. **Post-Training Quantization** (PTQ) quantizes a pre-trained network using a small calibration dataset and is thus relatively simple to implement (Nagel et al., 2020; Li et al., 2021; Frantar & Alistarh, 2022; Zhao et al., 2019; Cai et al., 2020; Nagel et al., 2019; Shao et al.,

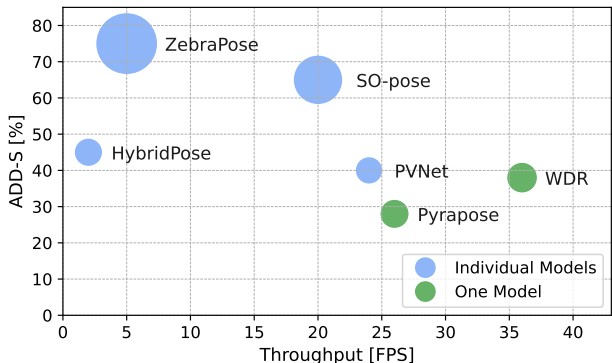

Figure 2: **Performance Comparison on Occluded-LINEMOD.** The marker size is proportional to the memory footprint. *Individual Models* refers to methods training one model for each object. *One Model* refers to methods training a single model for all objects.

2024; Lin et al., 2024; Chee et al., 2023). **Quantization-Aware Training** (QAT) retrains the network during the quantization process and thus better preserves the model's full-precision accuracy. QAT methods include uniform QAT (Esser et al., 2020; Zhou et al., 2017; Bhalgat et al., 2020; Yamamoto, 2021) and mixed-precision QAT (Cai & Vasconcelos, 2020; Dong et al., 2020; Tang et al., 2022; Dong et al., 2019; Chen et al., 2021; Yao et al., 2020). As accuracy can be critical in robotics applications relying on 6D object pose estimation, we focus on QAT.

**Uniform QAT** methods quantize every layer of the network to the same precision. In Incremental Network Quantization (INQ) (Zhou et al., 2017), this is achieved by quantizing a uniform fraction of each layers' weights at a time and continuing training until the next quantization step. Quantization can be achieved in a structured manner, where entire kernels are quantized at once, or in an unstructured manner. In contrast to INQ, Learned Step-size Quantization (LSQ) (Esser et al., 2020) quantizes the entire network in a single action. To this end, LSQ treats the quantization step-size as a learnable parameter. The method then alternates between updating the weights and the step-size parameter. Group-Wise Quantization (GWQ) (Yang et al., 2023) shows comparable accuracy while claiming reduced computational costs at inference time by assigning the same scale factor to different layers in the network.

Conversely, **Mixed-precision QAT** methods treat each network layer uniquely, aiming to determine the appropriate bit precision for each one. In HAWQ (Dong et al., 2019; 2020; Yao et al., 2020), the network weights' Hessian is leveraged to assign bit precisions proportional to the gradients. In (Cai & Vasconcelos, 2020), the mixed precision is taken even further by applying a different precision to different kernels within a single channel. Finally, NIPQ(Shin et al., 2023) replace the standard straight-through estimator ubiquitous in quantization with a proxy, allowing the quantization truncation and scaling hyper-parameters to be included in the gradient descent. Mixed-precision QAT is a challenging task; existing methods remain computationally expensive for modern deep network architectures.

## 2.3  Quantization and Modular Deep Learning

In recent years, deep network architectures have increasingly followed a modular paradigm, owing to its advantages in model design and training efficiency (Jacobs et al., 1991; Pfeiffer et al., 2023; Ansell et al., 2021; Hu et al., 2021a; Pfeiffer et al., 2020). This approach leverages reusable modules, amplifying the flexibility and adaptability of neural networks and fostering parameter-efficient fine-tuning. In the quantization domain, several studies have also underscored the importance of selecting the appropriate granularity to bolster model generalization (Li et al., 2021) and enhance training stability (Zhang et al., 2023).

Furthermore, in the 6D object pose estimation domain, the demonstration of applying quantization remains limited, with none of the aforementioned quantization techniques addressing the specific task or underlying network architecture. CA-SpaceNet (Wang et al., 2022) applied LSQ quantization to their proposed architecture using three different quantization configurations. However they do so homogenously, not accounting for the varying sensitivity to quantization throughout the network.

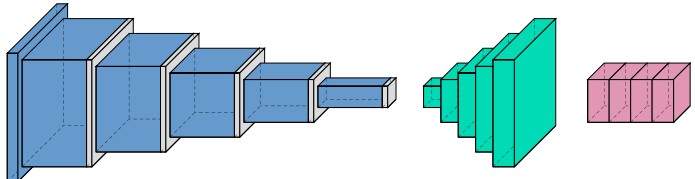

Figure 3: **Representative 6D Object Pose Estimation Network** with $K = 3$ modules. From left to right, we denote them as *backbone*, *feature aggregation*, and *heads*.

To our knowledge, no existing research has advocated a systematic methodology for executing modular quantization-aware training, let alone studied the impact of quantization order on modular architectures. Thus, in this work, we aim to bridge this gap by introducing a comprehensive methodology for modular QAT, tailored but not limited to 6D object pose estimation.

## 3 Method

In this section, we first describe the general type of network architecture we consider for compact 6D object pose estimation and then introduce our Modular Quantization-Aware Training (MQAT) method.

### 3.1 Modular Architecture for 6D Pose Estimation

As discussed in section 2, multi-stage networks (Su et al., 2022; Di et al., 2021; Wang et al., 2021; Labbé et al., 2020; Li et al., 2019b) tend to induce large memory footprints and large latencies, thus making poor candidates for hardware-restricted applications. This is further evidenced in fig. 2, where we compare a number of 6D object pose estimation architectures' memory footprint, throughput and accuracy on the O-LINEMOD dataset. While demonstrating admirable accuracy, the size and latency of ZebraPose (Su et al., 2022) and SO-pose (Di et al., 2021) preclude their inclusion in hardware-restricted platforms. On this basis, we therefore focus on single-stage[1] 6D object pose estimation networks (Thalhammer et al., 2021; Wang et al., 2022; Tekin et al., 2018; Peng et al., 2019; Hu et al., 2021b).

In general, 6D pose estimation networks contain multiple modules, with each module performing a distinct task. A typical encoder-decoder-head 6D pose estimation architecture is illustrated in fig. 3. Explicitly:

1. A *backbone* or *encoder* module extracts features from an image at different depths. Common module examples include ResNets, DarkNets and MobileNets.

2. A *feature aggregator* or *decoder* aggregates or interprets the latent feature space. Common module examples include feature pyramid networks, attention layers, and simple upscaling CNNs (i.e., latter half of U-Nets).

3. A *head* module performs specific tasks or regresses specific quantities such as location vectors and correspondences. Commonly correspondence and location heads are fully-connected layers but may also be attention-based or CNNs for other tasks such as image segmentation.

Architectures for various tasks similarly consist of clearly identifiable modules. Therefore, we also demonstrate the effectiveness of our proposed MQAT method for object detection, as detailed in the Appendix.

### 3.2 MQAT Algorithm

We introduce MQAT in algorithm 1 where we have also defined the notations. We begin by introducing the critical components for our method.

**Quantization Baseline** (*Lines 1:7*). MQAT starts with the aggressive quantization of individual modules to establish favourable bit precisions. For a modular network with $K$ modules, we conduct $K$ independent

---

[1]For completeness, we also demonstrate that our quantization method applies to two-stage networks (e.g., ZebraPose).

---

**Algorithm 1** MQAT Algorithm

---

**Input:** Training data, Quantization $Q$, Model $M$ with modules
$\mathbf{B} = [B_1, B_2, ..., B_K]$, Bit-width search $q = [2, 4, 8...]$, Accuracy
metric $ac(M)$

1: **for** $k = 1, 2, ..K$ **do**
2:      $M_k^2 \leftarrow Q_k^2(M)$
3:      **if** $ac(M_k^2) > ac(M) \wedge ac(M_k^2) > ac(M_{best})$ **then**
4:          $M_{best} \leftarrow M_k^2$
5:          $k_{best} \leftarrow k$
6:      **end if**
7: **end for**
8: $flow = [\,]$
9: **if** $M_{best}$ **then**
10:      $M \leftarrow M_{best}$
11:      $flow \leftarrow index(SORT\_SIZE(\mathbf{B}))$
12:      $flow.pop(k_{best})$
13: **else**
14:      $flow \leftarrow index(SORT\_SIZE(\mathbf{B}))$
15: **end if**
16: $\rho \leftarrow ILP(M, q)$
17: $i \leftarrow 0$
18: **while** $i < len(flow)$ **do**
19:      $M^Q \leftarrow Q_{flow(i)}^{\rho(i)}(M)$
20:      retrain$(M^Q)$
21: **end while**
**Output:** Modular quantized model $(M^Q)$.

**Variables:**

| | |
|---|---|
| $flow$ | Sequence of quantization order of modules. |
| $k_{best}$ | Index of the quantized module which increased performance. |
| $K$ | Number of modules. |
| $M_{best}$ | Highest accuracy model containing a quantized module. |
| $M_k^q$ | Model with only module $k$ quantized to $q$ bits. |
| $M^Q$ | Model with modules quantized to different bit precisions. |
| $\rho$ | List of bit precisions for each module. |
| $Q_c^j$ | Quantization applied to module $B_c$ with $j$-bits. |
| $SORT\_SIZE(B)$ | Sort list of modules $\mathbf{B}$ based on their size. |

---

2 bit quantizations for each module, $B_k$. The module is retained as quantized if it results in an improved accuracy for the entire model, (i.e., $M_k^q$ outperforms $M$). This establishes a new baseline from which the remaining module bit precision optimization can be performed.

**Quantization Flow** (*Lines 8:15*). The sequence of module quantization —the *quantization flow*— is critical as quantizing the modules of a network is not commutative; we prioritize starting with modules that do not compromise accuracy, as quantization errors introduced early on are typically not mitigated by later steps. Moreover, we also observe that quantization-related noise can lead to weight instability (Défossez et al., 2022; Shin et al., 2023; Peters et al., 2023), hindering the performance of the quantized network. If no quantized module yielded an improved accuracy in the quantization baseline step, we proceed with quantizing the module with the lowest number of parameters first. Modules with a greater number of parameters will have more flexibility to adapt to aggressive quantization. The output of this algorithmic step is the recommended *quantization flow*, which is then passed to the next step.

**Optimal Bit Precision** (*Lines 16:21*). In MQAT, the optimal bit precision for each module (excepting a quantization baseline identified $M_{best}$, $k_{best}$) is ascertained through a process of constrained optimization, *Integer Linear Programming* (ILP), drawing inspiration from Yao et al. (2020). However, our methodology distinguishes itself by offering a lower degree of granularity. Central to our strategy is the uniform quantization of all layers within a given module to an identical bit-width. This design choice not only simplifies the computational complexity but also significantly enhances the hardware compatibility, an essential consideration for efficient real-world deployment.

To achieve this, we introduce the *importance metric* for modular quantization. This metric is conceptualized as the product of two factors: the sensitivity metric, $\lambda$, for each layer in a module which is computed by a similar approach to that in Dong et al. (2020), and the quantization weight error. The latter is calculated as the squared 2-norm difference between the quantized and full precision weights. Therefore, the importance metric is given by

$$\Omega_k = \frac{1}{L_k} \sum_{i=1}^{L_k} \frac{\lambda_i}{N_i} |Q(W_i) - W_i|_2^2, \tag{1}$$

with

     $k$   the $k$-th module in the modular network;

$i$   the $i$-th layer within module $k$;

$L_k$   the total number of layers in module $k$;

$N_i$   the number of parameters in the $i$-th layer of module $k$;

$\lambda_i$   the sensitivity of the $i$-th layer in module $k$;

$Q$   the quantization operation;

$W_i$   the weights of the $i$-th layer in module $k$.

Finding the optimal bit precisions using ILP is formulated as

$$\min_{\{\rho_k\}_{k=1}^{K}} \sum_{k=1}^{K} \Omega_k^{(\rho_k)},$$
$$\text{subject to} \quad \sum_{k=1}^{K} S_k^{(\rho_k)} \leq \frac{\text{Full Precision Model Size}}{\text{Compression factor}} \ . \tag{2}$$

In this formulation, $\rho_k$ denotes the bit-width for the $k^{th}$ module; $S_k$ denotes the model size of module $k$; and $K$ represents the total number of layers.

## 4  Experiments

Historically, quantization and other compression methods have been used to exercise a trade-off between inference accuracy and deployment feasibility, particularly in resource-constrained circumstances. In the following sections, we will show that our approach may yield a significant inference accuracy improvement during compression.

We first introduce the datasets and metrics used for evaluation. Then, we present ablation studies to explore the properties of MQAT; this result is directly compared to uniform and mixed QAT methods. Finally, we demonstrate the generality of our method applied to different datasets, architectures, and QAT methods.

### 4.1  Datasets and Metrics

The LINEMOD (LM) and Occluded-LINEMOD (LM-O) datasets are standard BOP benchmark datasets for evaluating 6D object pose estimation methods, where the LM dataset contains 13 different sequences consisting of ground-truth poses for a single object. LM-O extends LM by including occlusions. Similar to GDR-Net (Wang et al., 2021), we utilize 15% of the images for training. For both datasets, additional rendered images are used during training (Wang et al., 2021; Peng et al., 2019). Similarly to previous works, we use the ADD and ADI/ADD-S error metrics[2] (Hodaň et al., 2016) expressed as

$$e_{ADD}(P^{est}, P^{gt}) = \frac{1}{V} \sum_{i=1}^{V} \|P_i^{est} - P_i^{gt}\|_2 \ , \tag{3}$$

$$e_{ADI}(P^{est}, P^{gt}) = \frac{1}{V} \sum_{i=1}^{V} \min_{j \in [1,V]} \|P_i^{est} - P_j^{gt}\|_2 \ , \tag{4}$$

where $P_i^{est}$ and $P_i^{gt}$ denote the $i^{th}$ vertex of the 3D mesh after transformation with the predicted and ground-truth pose, respectively. We then report the accuracy using the *ADD-0.1d* and *ADD-0.5d* metrics, which encode the proportion of samples for which $e_{ADD}$ is less than 10% and 50% of the object diameter. Similarly, we report *ADI-0.1d* and *ADI-0.5d* metrics for swisscube dataset to be consistent with (Hu et al., 2021b).

While LM and LM-O present their own set of challenges, their scope is restricted to household objects, consistently illuminated without significant depth variations. Conversely, the SwissCube dataset (Hu et al.,

---

[2]The ADI/ADD-S metric was proposed for image sets with *indistinguishable* poses. ADI is used to compare fairly with previous works. (Hu et al., 2021b)

| Combination | | | ADI |
|---|---|---|---|
| H | F | B | |
| 32 | 32 | 32 | 78.8 |
| 2 | 32 | 32 | 69.1 |
| 32 | 2 | 32 | 83.8 |
| 32 | 32 | 2 | 67.8 |
| 2 | 2 | 32 | 70.0 |
| 2 | 32 | 2 | 56.3 |
| 32 | 2 | 2 | 69.4 |
| 2 | 2 | 2 | 45.1 |

Figure 4: **A heuristic search of quantization flow sequences** to demonstrate quantization flow optimality for K=3 module WDR. This corresponds to lines 1-7 in Algo.1.

| MQAT | ADI | |
|---|---|---|
| First Quantized Module | 0.1d | 0.5d |
| Full Precision | 78.79 | 98.98 |
| Backbone | 69.08 | 96.79 |
| Head | 67.84 | 98.12 |
| Feature Pyramid Network (FPN) | **83.8** | **99.4** |

Table 1: Effect of Starting MQAT with Different Modules.

2021b) embodies a challenging scenario for 6D object pose estimation in space, incorporating large scale variations, diverse lighting conditions, and variable backgrounds. To remain consistent with previous works, we use the same training setup and metric as Wang et al. (2022).

## 4.2 Implementation Details

We use PyTorch to implement our method. For the retraining of our partially quantized pretrained network, we employ an SGD optimizer with a base learning rate of `1e-2`. It is common practice for quantization algorithms to start with a pre-trained model (Esser et al., 2020; Dong et al., 2019; Zhou et al., 2017); we similarly do so here. We employed 30 epochs to identify the starting module at 2bits (Alg. 1 Lines 1-7) and then trained 30 epochs per module (Alg. 1 Lines 18-21). Therefore MQAT training time will scale linearly with additional modules. Specifically, WDR with M=3 results in 120 epochs and ZebraPose with M=2 results in 90 epochs. We use the same training time respectively for LSQ, INQ, HAWQ for a fair comparison. For all experiments, we use a batch size of 8 and employ a hand-crafted learning scheduler which decreases the learning rate at regular intervals by a factor of 10 and increases it again when we quantize a module with INQ[3]. However, when we quantize our modules using LSQ, the learning rate factor is not increased, only decreased by factors of 10. We use a $512 \times 512$ resolution input for the SwissCube dataset and $640 \times 480$ for LM and LM-O as in Peng et al. (2019).

## 4.3 MQAT Paradigm Studies

In this section, we conduct comprehensive studies using SwissCube and demonstrate the superior performance of our approach over conventional QAT ones.

### 4.3.1 MQAT Quantization Flow

We first perform an ablation study to validate the recommended quantization flow's sequence optimality. As discussed in section 3.2, the module quantization sequence is not commutative. Using WDR, we perform aggressive quantization to every combination of modules in the network. This is an $O(2^K)$ search; this results in eight module quantization combinations for a network with $K = 3$ modules. The results are visualized in fig. 4. The backbone and head modules exhibit greater sensitivity to aggressive quantization. Conversely, the accuracy of the network is enhanced when using 2 bit quantization on the Feature Pyramid Network (FPN) module only. No other combination of module quantizations yields an accuracy increase. This further emphasizes the importance of carefully selecting a module quantization flow.

We additionally perform ablation studies on the optimal order (i.e., flow) of module quantization. We begin by quantizing different modules first, instead of the FPN. Table 1 shows the results of both the head and backbone modules when they are the first module quantized. We observe that the inference accuracy decreases dramatically for both cases. No combination of module flow or bit precision schedule is able to

---

[3]The learning rate and quantization schedulers are provided in the appendix.

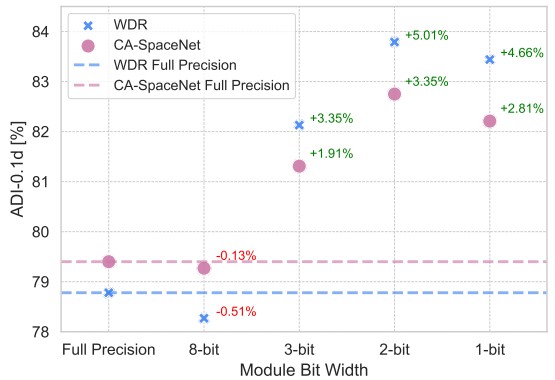 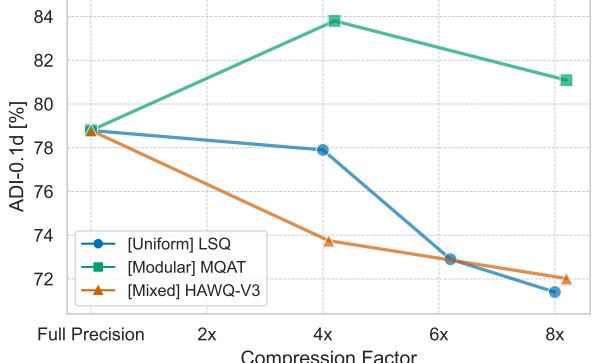

Figure 5: **Ablation study on the FPN bit-width**. We compare the performance by varying the bit-width of the feature aggregation module in each model.

Figure 6: Comparison between our proposed paradigm MQAT, uniform QAT (LSQ), and layer-wise mixed-precision QAT (HAWQ-V3).

Table 2: **Comparison with the state-of-the-art on SwissCube.** We report ADI-0.1d scores for three different depth ranges. A * indicates applying MQAT with 2-bit precision FPN to the model.

| Network | Near | Medium | Far | All |
|---|---|---|---|---|
| SegDriven-Z (Hu et al., 2019) | 41.1 | 22.9 | 7.1 | 21.8 |
| DLR (Chen et al., 2019) | 52.6 | 45.4 | 29.4 | 43.2 |
| CA-SpaceNet | 91.0 | 86.3 | 61.7 | 79.4 |
| CA-SpaceNet* | 95.5 | 90.7 | 66.2 | 82.7 |
| WDR | 92.4 | 84.2 | 61.3 | 78.8 |
| WDR* | **96.1** | **91.5** | **68.2** | **83.8** |

recover the inference accuracy after it was lost. The 2 bit aggressive FPN quantization yields improved accuracy only when the FPN is quantized first.

### 4.3.2 Quantized FPN Sensitivity Study.

To expand upon the 2 bit FPN accuracy enhancement, we perform a higher granularity bit-precision search on the FPN module. Again, the FPN module was quantized, but to five different bit-widths for comparison; the results are presented in fig. 5. The accuracy of two full-precision networks, WDR (Hu et al., 2021b) and CA-SpaceNet (Wang et al., 2022), are shown with dashed lines. The highest accuracy is achieved with a 2 bit or ternary $\{-1, 0, 1\}$ FPN. Further pushing the FPN to binary weights $\{-1, 1\}$ slightly reduces the accuracy, but maintains a significant improvement over both baselines[4].

In the context of our study, as depicted in fig. 5, our findings support the premise that varying bit-precisions across different modules within QAT, such as the FPN, can significantly influence overall performance. This can be attributed to a redistribution of computes or inherent regularization effects on the modules. The improvements in performance reinforce the potential of MQAT in enhancing model generalizability.

### 4.3.3 MQAT Compared to Uniform and Mixed QAT.

For direct comparison, we apply three different quantization paradigms. Starting from a full precision WDR network, we apply a uniform QAT method, LSQ (Esser et al., 2020), a mixed-precision QAT method, HAWQ-V3 (Yao et al., 2020), and finally our proposed MQAT method with increasing compression factors. The results are provided in fig. 6. Again, MQAT demonstrates a significant accuracy improvement comparing to other methods while sustaining the requested compression factor; it is the only quantization approach to show an increase in inference accuracy during compression.

---

[4]For the interested reader, the FPN layer-wise ADI-0.1d accuracies are provided in the appendix.

Table 3: **Quantized FPN in WDR network on different datasets.** We report ADI for Swisscube and ADD for LM/LM-O, with the quantization flow F→H→B.

| MQAT Mode | Bit-Precisions | SwissCube | | LM | | LM-O | |
|---|---|---|---|---|---|---|---|
| | | 0.1d | 0.5d | 0.1d | 0.5d | 0.1d | 0.5d |
| Full precision | Full precision | 78.8 | 98.9 | 56.1 | 99.1 | 37.8 | 85.2 |
| $MQAT_{LSQ}$ | 8-2-8 | 83.4 | 99.3 | 63.5 | 99.2 | 39.8 | 86.4 |
| $MQAT_{INQ}$ | 8-2-8 | **83.7** | **99.4** | **63.9** | **99.5** | **40.2** | **86.7** |

## 4.4 MQAT Generality

Finally, we demonstrate the generality of MQAT to different datasets, QAT methods, and network architectures. Additionally, we showcase its applicability to a different task: object detection.[5]

### 4.4.1 Dataset and QAT Generality

As discussed in section 4.1, the image domains of LM, LM-O and SwissCube are vastly different. The full precision and MQAT quantized models results for all three datasets are shown in table 3. MQAT demonstrates an accuracy improvement in all datasets. We use the ADI metric for evaluation on the SwissCube dataset as in (Hu et al., 2021b; Wang et al., 2022), while we use the ADD metric for LM and LM-O as used by (Wang et al., 2021; Su et al., 2022; Thalhammer et al., 2021; Labbé et al., 2020; Peng et al., 2019).

Accuracy improvements of 5.0%, 7.8% and 2.4% are demonstrated on SwissCube, LM and LM-O, respectively, when MQAT with INQ is utilized. Replacing INQ with LSQ yields accuracy improvements of 4.6%, 7.5% and 2.0%, respectively. This evidences that the performance enhancement is independent of the dataset domain and the applied QAT method.

As discussed in section 3.2 and section 4.3.1, it is difficult to recover accuracy once it is lost during quantization. To this end, since INQ (Zhou et al., 2017) quantizes only a fraction of the network at once, it follows that the remaining unquantized portion of the network is left flexible to adapt to aggressive quantization. Conversely, LSQ (Esser et al., 2020) quantizes the entire network in a single step; no fraction of the network is left unperturbed. Consequently, INQ demonstrates superior results in table 3. While any QAT method may be used, we recommend partnering MQAT with INQ for optimal aggressive quantization results.

### 4.4.2 Architecture Generality

In table 2, we compare several single-stage PnP architectures on the SwissCube dataset. To demonstrate the generality of our performance enhancement, we apply MQAT to aggressively quantize the FPN of both CA-SpaceNet (Wang et al., 2022) and WDR (Hu et al., 2021b). We demonstrate an accuracy improvement of 4.5%, 4.4% and 4.5% for Near, Medium and Far images, respectively, on CA-SpaceNet, resulting in a total testing set accuracy improvement of 3.3%. Recall the already presented total testing set accuracy improvement of 5.0% for WDR. Previously, the full precision CA-SpaceNet had shown a performance improvement over the full precision WDR, but WDR sees greater gains from the application of MQAT.

In addition, (Wang et al., 2022) published accuracy results for a uniform QAT quantized CA-Space network, shared in table 4. Specifically, CA-SpaceNet explored three quantization modes (B, BF and BFH). These correspond to quantizing the backbone, quantizing the backbone and FPN (paired), and quantizing the whole network (uniformly), respectively.

Finally, we evaluate MQAT on a multi-stage network architecture. Specifically, in Table 5, we demonstrate the performance of our method on the state-of-the-art 6D object pose estimation network, the two-stage ZebraPose network (Su et al., 2022). Note that, when quantized using MQAT, the model's performance comes close to that of its full-precision counterpart while being more than four times smaller. Furthermore, MQAT outperforms the state-of-the-art HAWQ-V3 (Yao et al., 2020) by ∼2.6%, with the added advantage of further network compression. To further demonstrate the efficacy, we quantize another two stage network i.e

---

[5]The results for the object detection task are provided in the appendix.

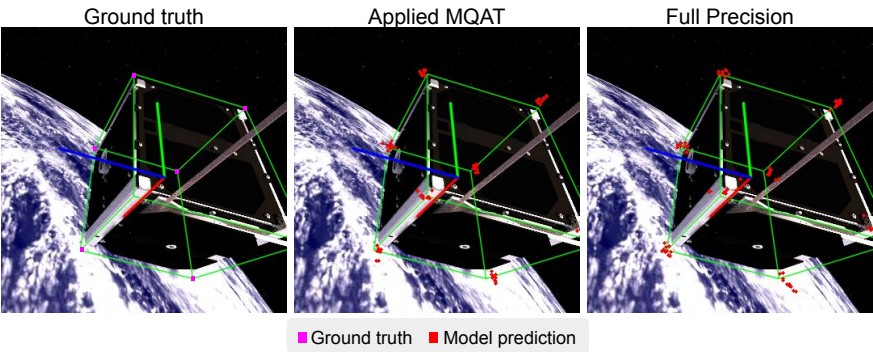

Figure 7: **Visualization of predictions.** Comparison between our proposed MQAT paradigm, and full-precision network. The model applying MQAT yields predictions that are on par with, or more concentrated than its full-precision counterpart.

Table 4: **CA-SpaceNet Published Quantization vs MQAT.** We report ADI scores on the SwissCube dataset sorted by the compression factor of the network, for MQAT methods, we use quantization flow F→H→B.

| Quantization Method | ADI-0.1d | Compression | Bit-Precisions (B-F-H) |
|---|---|---|---|
| LSQ | 79.4 | 1× | 32-32-32 |
| LSQ B | 76.2 | 2.2× | 8-32-32 |
| LSQ BF | 75.0 | 3.2× | 8-8-32 |
| LSQ BFH | 74.7 | 4.0× | 8-8-8 |
| **MQAT (Ours)** | **82.7** | 4.7× | 8-2-8 |
| LSQ B | 75.1 | 2.9× | 3-32-32 |
| LSQ BF | 74.5 | 5.9× | 3-3-32 |
| **MQAT (Ours)** | **80.2** | 8.2× | 4-2-4 |
| LSQ BFH | 68.7 | 10.6× | 3-3-3 |

GDR-Net with existing uniform and mixed-precision methods, and MQAT. We employed ADD-0.1d metric for 6D object pose evaluation for O-Linemod dataset as Wang et al. (2021). It is evident from table 6 that MQAT outperforms both LSQ (Esser et al., 2020) and HAWQ-V3 (Yao et al., 2020) even with slightly more compressed network.

As we demonstrated in section 4.3.1, quantizing network modules all together greatly reduces inference accuracy as the smaller unquantized fraction of the network is not able to adapt to the quantization. Additionally, quantizing from backbone to head does not consider the sensitivity of the network modules to quantization. As a final note, CA-SpaceNet does not quantize the first and last layer in any quantization mode. In contrast, MQAT quantizes the entire network.

## 5    Limitations and Discussion

***Module Granularity.***    As conclusively demonstrated in section 4.3.1, MQAT exploits the modular structure of a network. Therefore, if the network does not contain distinct modules, MQAT simply converges to a uniform QAT methodology. In principle, MQAT can apply to any architectures with $K \geq 2$.

***Latency.***    Directly reporting latency measurements involves hardware deployment, which goes beyond the scope of this work. However, as shown in Yao et al. (2020), latency is directly related to the bit operations per second (BOPs). With lower-precision networks, both the model size and the BOPs are reduced by the same compression factor, which we provide in our experiments. Therefore, it is expected that MQAT would demonstrate a latency improvement proportional to the network compression factor.

Nonetheless, we can share some results running an MQAT quantized network on a CPU. Running WDR on our **Intel Core i7-9750H** CPU demonstrates a latency of **650ms** for full precision and **299ms** for our MQAT int8 quantized model. However this does not exploit the full benefits of our MQAT quantized model as all the CPU deployed parameters are stored at the highest common denominator of INT8 (whereas our model employs lower bit-widths for certain parameters).

Table 5: **Quantization of ZebraPose (Su et al., 2022).** We report ADD scores on the LM-O dataset and compare MQAT to mix-precision quantization (HAWQ-V3).

| Quantization Method | ADD 0.1d | Compression | Bit-Precisions | Quantization Flow |
|---|---|---|---|---|
| Full precision | 76.90 | 1× | Full precision | N/A |
| HAWQ-V3 (Yao et al., 2020) | 71.11 | 4× | Mixed (layer-wise) | N/A |
| HAWQ-V3 (Yao et al., 2020) | 69.87 | 4.60× | Mixed (layer-wise) | N/A |
| **MQAT $K = 2$ (ours)** | **72.54** | **4.62×** | 8-4 (B-D) | D → B |

Table 6: **Quantization of GDR-Net (Wang et al., 2021).** We report ADD scores on the LM-O dataset and compare MQAT to uniform (LSQ) and mix-precision quantization (HAWQ-V3). B, R and P indicates *Backbone*, *Rotation Head* and *PnP-Patch* modules.

| Quantization Method | ADD 0.1d | Compression | Bit-Precisions | Quantization Flow |
|---|---|---|---|---|
| Full precision | 56.1 | 1× | Full precision | N/A |
| LSQ (Esser et al., 2020) | 50.7 | 4.57× | Uniform(7-bit) | N/A |
| HAWQ-V3 (Yao et al., 2020) | 50.3 | 4.9× | Mixed (layer-wise) | N/A |
| **MQAT (ours)** | **51.8** | **4.97×** | 8-4-4 (B-R-P) | R → P → B |

***Recommended Quantization Flow Optimality.*** A major contribution of this paper is the identification of the existence of an asynchronous optimal quantization sequence or flow. In section 3.2 we provide a recommended quantization flow and exhaustively demonstrate its optimality for $K = 3$ in fig. 4. However, the recommended quantization flow's optimality has yet to be analytically proven for all combinations of networks and number of modules, $K$.

## 6 Conclusion

We have introduced Modular Quantization-Aware Training (MQAT) for networks that exhibit a modular structure, such as 6D object pose estimation architectures. Our approach builds on the intuition that the individual modules of such networks are unique, and thus should be quantized uniquely while heeding an optimal quantization order. Our extensive experiments on different datasets and network architectures, and in conjunction with different quantization methods, conclusively demonstrate that MQAT outperforms uniform and mixed-precision quantization methods at various compression factors. Moreover, we have shown that it can even enhance network performance. In particular, aggressive quantization of the network FPN resulted in 7.8% and 2.4% test set accuracy improvements over the full-precision network on LINEMOD and Occluded-LINEMOD, respectively. In the future, we will investigate the applicability of MQAT to tasks other than 6D object pose estimation and another potential follow up would be to apply MQAT to architectures with even more modules or to instead classify large modules as two or more modules for the purposes of MQAT.

## 7 Acknowledgements

We thank Ziqi Zhao for assisting us with the experiments on Multi-stage architecture. This project has received funding from the European Union's Horizon 2020 research and innovation programme under the Marie Skłodowska-Curie grant agreement No. 945363. Moreover, this work was funded in part by the Swiss National Science Foundation and the Swiss Innovation Agency (Innosuisse) via the BRIDGE Discovery grant No. 194729.

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

# A  Appendix

## A.1  Multi-scale Fusion Analysis of MQAT Applied to WDR (Hu et al., 2021b)

In this section we study the inference performance of our MQAT when applied to WDR, our primary focus is to understand the impact of MQAT on the multi-scale fusion inference of WDR, particularly examining changes in performance across individual layers of FPN module in WDR. To this end, we applied MQAT to the WDR architecture and conducted a layer-by-layer performance analysis of FPN module similar to the original WDR paper. The results are presented in Table 7. A notable observation from this analysis is the overall enhancement in performance across most layers and the improvement is consistent across various scales and depth ranges. Particularly, Layer 1 exhibits a significant performance boost, especially for objects classified as *Far*. This layer-specific insight underscores the effectiveness of MQAT in optimizing the WDR network at a granular level.

| Layer | Near | Medium | Far | All |
|:-----:|:----:|:------:|:---:|:---:|
| 1 | 13.6 (+0.8) | 83.8 (+1.0) | 55.1 (+6.2) | 52.9 (+3.0) |
| 2 | 13.6 (+2.6) | 83.8 (+1.3) | 55.1 (+0.3) | 54.1 (+1.3) |
| 3 | 16.3 (-0.5) | 77.8 (+1.6) | 17.1 (-1.2) | 37.2 (-0.1) |
| 4 | 13.0 (+0.7) | 0.41 (+0.7) | 0 (0) | 3.8 (+0.4) |

Table 7: **Layer-wise 6D Pose Validation Results with MQAT on WDR's FPN.** ADI-0.1d are reported for each layer. Notable performance enhancements, particularly in Layer 1 for *Far* objects, illustrate the effective quantization of WDR by MQAT.

## A.2  Generality of MQAT in Object Detection Problem

In our study, while the primary focus is on 6D pose estimation, where our method's efficacy is already demonstrated, we further extend our evaluation to object detection tasks. This extension is aimed at underscoring the generality of our approach.

To this end, we applied our quantization technique to the Faster R-CNN network, which utilizes a ResNet-50 backbone, a widely recognized model in object detection tasks. Our evaluation was conducted on the comprehensive COCO dataset, a benchmark for object detection.

| Network | QAT Method | mAP | Compression |
|:-------:|:----------:|:---:|:-----------:|
| FasterRCNN | Full Precision | 37.9 | 1x |
|  | FQN | 32.4 | 8x |
|  | INQ | 33.4 | 8x |
|  | LSQ | 33.8 | 8x |
|  | **MQAT (ours)** | **35.1** | **8x** |
| EfficientDet-D0 | Full Precision | 33.16 | 1x |
|  | N2UQ | 20.11 | 10x |
|  | **MQAT (ours)** | **21.67** | **10x** |

Table 8: **Quantization for Object Detection.** We evaluate the given networks on COCO dataset and report mAP.

The results of this experiment are summarized in Table 8. Here, our method, denoted as MQAT, is compared against other quantization approaches: INQ (Zhou et al., 2017), LSQ (Esser et al., 2020), and the FQN (Li et al., 2019a) method, which has been specifically tailored for object detection tasks. The comparison reveals

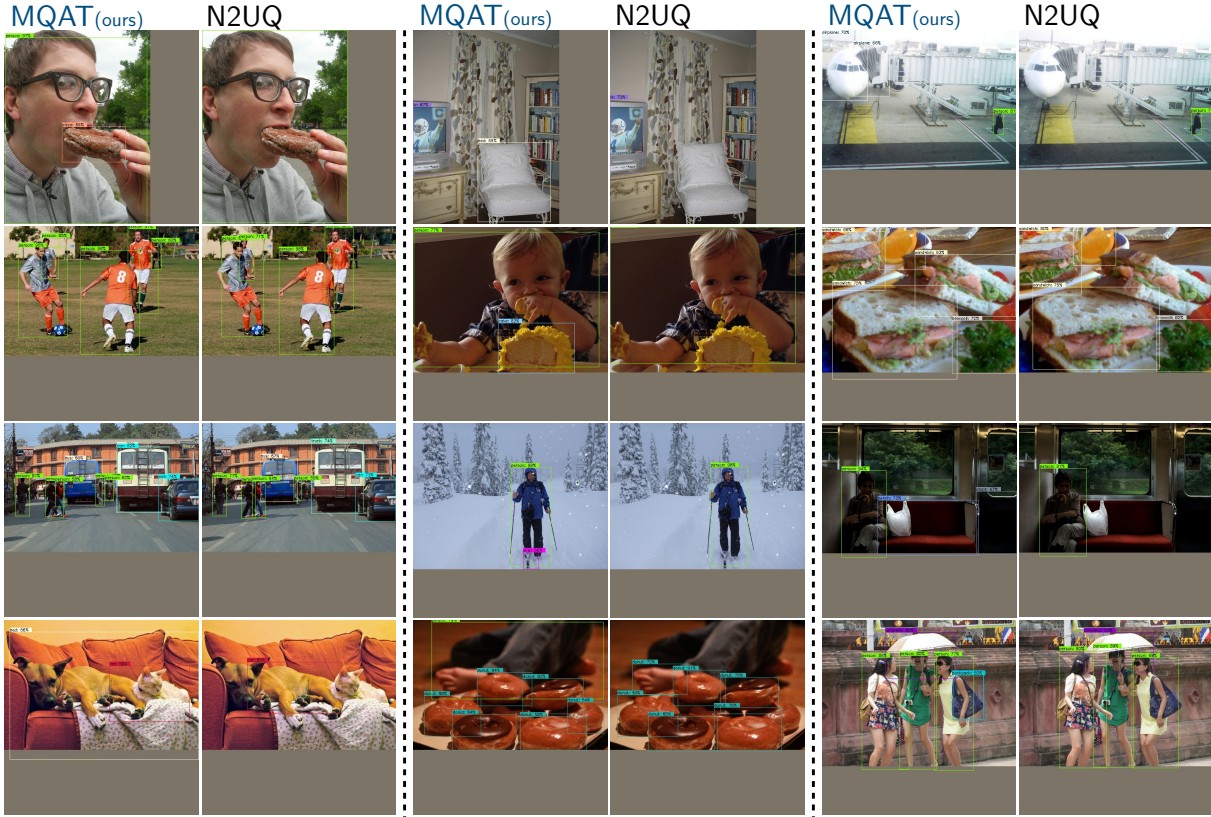

Figure 8: **Visualization of the Difference in Object Detection Performance** on MSCOCO (Lin et al., 2014) between N2UQ and MQAT at the same compression ratio.

that MQAT not only adapts well to a different task domain but also achieves superior performance over these established Quantization-Aware Training techniques. This underlines the adaptability and robustness of our approach, extending its potential applications beyond 6D pose estimation to broader areas within computer vision.

Moreover, we also created a baseline for the network, EfficientDet (Tan et al., 2020) by quantizing it with a recent quantization method: Non-Uniform to Uniform Quantization (N2UQ) (Liu et al., 2022)). This comparison is crucial to validate the effectiveness of MQAT across various modular architectures and quantization methods. Remarkably, our MQAT approach demonstrated a performance improvement of approximately 1.6% over N2UQ as shown in both Table 8. and fig. 8. This enhancement was observed under comparable compression ratios and identical training durations, further substantiating the superiority of MQAT in terms of efficiency and effectiveness.

## A.3 More Results on Speed+ Dataset (Park et al., 2022)

### A.3.1 Overview of Speed+ Dataset

The Next Generation Spacecraft Pose Estimation Dataset (Speed+) addresses the the domain gap challenge in spacecraft pose estimation. it encompasses 60,000 synthetic images, divided into an 80:20 train-to-validation ratio. The test set comprises of 9,531 Hardware-In-the-Loop images of the half-scale mockup model of the Tango spacecraft.

### A.3.2 Results on Speed+ Dataset

Our method was further tested on the Speed+ dataset. The WDR network was quantized with our proposed MQAT and the results are shown in Table 9, where we exclusively assess the model on the validation dataset.

| Network | ADI-0.1d |
|---|---|
| Full precision | 96.2 |
| $\text{MQAT}_{(8-2-8)}$ | **99.1** |

Table 9: **Evaluation of MQAT on Speed+ dataset.**

## A.4 More Implementation Details

### A.4.1 LR Schedule for INQ (Zhou et al., 2017)

As mentioned in our experiments section, we employ a SGD optimizer with a base learning rate ($lr_b$) of $1e^{-2}$. We trained for 30 epochs with a batch size of 8. We created a hand-crafted learning scheduler which decreases the learning rate at regular intervals by a factor of 10 and increases it again when we quantize a module. The gamma ($\gamma$) and quantization fraction scheduler are shown in Table 10. The quantization fraction corresponds to the percentage of weights quantized at each epoch. At each epoch, the learning rate ($lr$) is computed as:

$$lr = lr_b * \gamma$$

| Epoch | Schedule | |
|---|---|---|
| | $\gamma$ | fraction |
| 0 | 1 | 0.2 |
| 3 | 0.1 | - |
| 5 | 1 | 0.4 |
| 7 | 0.1 | - |
| 9 | 1 | 0.6 |
| 11 | 0.1 | - |
| 13 | 1 | 0.8 |
| 15 | 0.1 | - |
| 17 | 1 | 0.9 |
| 19 | 0.1 | - |
| 21 | 1 | 0.95 |
| 23 | 0.1 | - |
| 25 | 1 | 0.975 |
| 27 | 0.1 | - |
| 29 | 1 | 1 |
| 30 | 0.1 | - |

Table 10: **Learning Rate Schedule**.

### A.4.2 LR Schedule for LSQ (Esser et al., 2020) and HAWQ-V3 (Yao et al., 2020)

As mentioned in our experiments section, we employ a SGD optimizer with a base learning rate ($lr_b$) of *1e-2*. However, we used a multi-step decay scheduler here. Learning rate was decreased by factors of 10 at epochs $\{10, 20, 25\}$ of the training. We trained for 30 epochs with a batch size of 8.

### A.5 Reproducibility

The source code will be provided publicly along with the details of the environments and dependencies. Moreover, we will provide instructions to reproduce the main results in the manuscript.

