# OpenReview forum: "Modular Quantization-Aware Training for 6D Object Pose Estimation"
_TMLR — Accepted by TMLR_

### Review · Reviewer_8Fjr · 2024-08-27

**Summary Of Contributions:**

The paper introduces Modular Quantization-Aware Training (MQAT), an approach to quantization-aware training (QAT) specifically designed for 6D object pose estimation networks. The key contribution of MQAT is its adaptive, mixed-precision quantization strategy that leverages the modular structure of these networks. Unlike traditional uniform or mixed-precision QAT methods, MQAT performs a gradated modular quantization sequence, determining module-specific bit precisions. This method results in quantized models that outperform state-of-the-art methods both in terms of accuracy and memory efficiency.

The authors validate their approach across multiple datasets and architectures, showing the applicability of MQAT. Additionally, they present ablation studies that highlight the impact of module-specific quantization order and bit precision on performance.

**Audience:**

Yes

**Broader Impact Concerns:**

N.A.

**Claims And Evidence:**

No

**Requested Changes:**

A brief introduction to the 6D object pose estimation task should be added to the paper.
The technical detail part of the paper should warrant the paper are replicable.

**Strengths And Weaknesses:**

Strength: The paper provides robust empirical evidence that MQAT consistently outperforms both uniform and mixed-precision QAT methods in terms of accuracy.

Weakness: The paper lacks sufficient technical details, particularly in the implementation section. For instance, it is unclear how long the models are trained and whether the baseline models are trained and tuned in a manner comparable to the models using the proposed approach. Additionally, the significance of the proposed modularization is not well-established. MQAT, as described, appears more similar to post-training quantization (PTQ) since it relies on a model trained with quantization awareness. Furthermore, the experiments exploring the optimal MQAT order are somewhat limited and could benefit from more comprehensive analysis.

---

> ### Author Response · Authors · 2024-09-15
>
> Thank you for your thoughtful feedback on our paper. Below we address the weaknesses and highlight the related changes in the revised manuscript.
>
> >**W1: Lack of Sufficient Technical Details (Training Time, Baseline Comparisons)**
>
> A: We understand the importance of providing a more comprehensive comparison and training details. We clarify that the baseline models were indeed trained and fine-tuned under similar settings, ensuring fair comparisons across methods. Due to the 12 page limit, training times for both MQAT and baseline models for WDR were originally provided in the Appendix. However, based on reviewer feedback, we agree these details should be prioritized in the main paper and have correspondingly updated section 4.2.
>
> For convenience, we are also happy to share the update here.
>
> “It is common practice for quantization algorithms to start with a pre-trained model [1, 2, 3]; we similarly do so here. We employed 30 epochs to identify the starting module at 2bits (Alg.1 Lines 1-7) and then 30 epochs per module (Alg.1 Lines 18-21). As MQAT assigns 30 epochs of training to each module, MQAT training time will scale linearly with additional modules. Specifically, WDR with M=3 results in 120 epochs and ZebraPose with M=2 results in 90 epochs. We use the same training time respectively for [1, 2, 3] for a fair comparison.”
>
> [1] HAWQV3: Dyadic Neural Network Quantization. ICML, 2020.\
> [2] Learned step size quantization. ICLR, 2020 \
> [3] Incremental Network Quantization: Towards lossless cnns with low-precision weights. ICLR, 2017
>
>
> >**W2: Significance of Modularization and MQAT appears similar to Post-training Quantization (PTQ)**
>
> Please note that all QAT methods typically begin with pre-trained models [1, 2, 3]. While PTQ quantizes a model after training (no quantization awareness), a QAT method applies training during the quantization process. For this reason, we compare against LSQ, INQ and HAWQ.
>
> Unique from all quantization approaches, MQAT identifies and exploits the modularity of recent deep network architectures (related work section 2.3). MQAT dynamically assigns different bit precisions to each module based on sensitivity and parameter count. This adaptiveness is not typical in PTQ or standard QAT.
>
> [1] HAWQV3: Dyadic Neural Network Quantization. ICML, 2020.\
> [2] Learned step size quantization. ICLR, 2020 \
> [3] Incremental Network Quantization: Towards lossless cnns with low-precision weights. ICLR, 2017
>
> >**W3: Limited Experiments on Optimal MQAT Order**
>
> We appreciate this feedback and acknowledge that exploring the optimal ordering further could provide additional insights. However, in the current version of the paper, we performed several experiments with different architectures and datasets to validate the ordering based on module sensitivity and parameter size can demonstrate better compression and accuracy. We additionally empirically proved our quantization order as optimal for the WDR case by performing a heuristic search of the module quantization order space in section 4.3.1. It is not feasible to perform this search across all architectures and datasets.
>
> >**Changes: A Brief Introduction to 6D Object Pose Estimation Task**
>
> Per your feedback we have included a definition of 6D pose estimation in the introduction.

---

> > ### Comment · Reviewer_8Fjr · 2024-09-20
> >
> > Thank you for taking the feedback into account. Overall your revisions greatly improved the paper's clarity and technical depth. I have no more furhter concerns.

---

### Review · Reviewer_Fo2q · 2024-09-09

**Summary Of Contributions:**

This work provides a new quantization-aware training technique that is suitable for 6D object pose estimation; it uses the modular structure of networks to assign different quantization bit-width to each module and used greedy type algorithm to determine an order of the quantizations of modules.  This work then present several experimental results that show advantages of the proposed approach over existing baseline approaches through careful comparisons.

**Audience:**

Yes

**Claims And Evidence:**

Yes

**Requested Changes:**

Please address the weakness I provided above.

**Strengths And Weaknesses:**

Strength:

The proposed approach seems very simple, but is shown to exhibit advantages over baselines and standard datasets; and the comparisons are rather careful.

Weakness:

Although the contents are interesting, it is a bit hard to see the overviews at first.

Major:
1. I think the main contributions of the proposed approach are (1) exploitation of module structure for QAT and (2) greedy type algorithm for determining the order.  And it is mentioned that the former converges to uniform QAT when there is no module structure.  I strongly recommend giving some tables in the introduction etc. to summarize comparisons of the difference of ways of quantization of this approach against existing ones.  Otherwise, it is a bit too hard to understand this aspect from the texts only.

2. Related to the above, the authors should clarify these two aspects of the proposed approach, i.e., the module structure and the way it determines the quantization order.  In the experiment, I could see the advantage for each in a separate manner somehow, however, these two are very distinct ideas and the authors should clarify them in a crystal clear manner.

3. It seems that the proposed approach involves large computations for deciding the order; if that is the case, presenting the comparison of computational burden and convincing us that it still is usable on a resource restricted hardware are important.

4. The authors mention that the proposed algorithm is for optimal order but also say that it is not proved optimal in the limitation section.  This should be clear in the main text; I.e, the proposed algorithm does not provide THE optimal order in general and is a heuristic search.

Overall, it seems that the way it presents the work is not really well structured and claims are rather scattered.  I would like to see very clear statements of the technical contributions (in addition to the ones in the bullet points in page 2) and claims associated with them so that I can see the clear connections between the claims and the experimental results.  (like, the authors also mention that the proposed approach is the only one that achieves accuracy increase via quantization.  These kinds of claims are scattered around the texts.)

Minor:
1. What type of experiments are used to show fig 1.?  Is it something like average result?
2. 2.1. instead proposed … instead of …   → maybe use one “instead”?
3. 3.2. … methods … approaches uniformly apply → delete methods or approaches
4. fig 4; color the first row or so to indicate that it is the baseline (i.e., full precision)
5.   4.4.2 we aggressively quantize the FPN of apply …  → fix typos
6. 5. existance → existence ?

---

> ### Author Response · Authors · 2024-09-15
>
> We thank the reviewer for their valuable feedback. We address their major concerns below and highlight the related changes in the revised manuscript.
>
> **W1: Comparison with other Methods in Tabular fashion**
>
> Thank you for the suggestion to include a table to help summarize comparisons of different quantization approaches, but we do not believe a table will be more explanatory than our current Figure 1. On the right side of Figure 1, there are three depictions of different quantization approaches for a 3 module network. In the case of uniform QAT, all modules are uniformly quantized. In mixed-precision QAT, layers are quantized to varying bit precisions regardless of their position. In contrast, MQAT applies quantization to network modules in a proposed order, with each module assigned the optimal bit precision. We have clarified this in an extended caption for Figure 1.
>
>
> >**W2: Clarification of “Modular Structure” and “Quantization Flow” aspects**
>
> As per your feedback, we are currently in the process of rewording the Section 1 contributions and Section 3.2 MQAT overview. We will upload the revisions shortly.
>
> >**W3: Comparison of Computational Burden**
>
> If this question is with regards to the computational burden during training, we have updated section 4.2 with a QAT method training time comparison. This data was originally in the appendix.
>
> If this question is with regards to the computational burden on hardware at inference we have CPU latency results. Running WDR on our Intel Core i7-9750H CPU demonstrates a latency of 650ms for full precision and 299ms for our MQAT quantized model. We have added this to the appendix A.4.1.
>
> We should also point out that theoretically, MQAT has a lower computational burden than other mixed-precision QATs such as HAWQ. Whereas HAWQ may require overhead bit precision alterations between each layer, MQAT will only change bit precisions between each module.
>
> >**W4: Clarification of quantization flow optimality.**
>
> A: We agree this point could be clarified. MQAT proposes a recommended quantization flow. We empirically demonstrate that the proposed quantization flow is optimal with a greedy heuristic search of the module ordering space in section 4.3.1 MQAT order. However, this empirical proof of optimality is only valid for the specific case of WDR. Thus we state in the limitations section that our recommended quantization flow has not (yet?) been proven optimal for all architectures. Per this feedback, we will rework the page 2 contributions for clarity and carefully curate our use of the term "optimal" in the remainder of the paper (particularly section 3). We will upload the revisions shortly.
>
> ***minor***:
> 1. The results presented in Figure 1 are not averaged. Rather, they represent a comparison of different quantization methods applied to the WDR network, evaluated using the Swisscube dataset. We have added this clarification to the figure caption for improved clarity.
>
> 2. We have revised the sentence for clarity by removing one "instead."
>
> 3. We have adjusted the phrasing to uniformly use "methods," ensuring consistency.
>
> 4. We have modified Figure 4 by coloring the first row to clearly indicate that it represents the baseline (i.e., full precision).
>
> 5. The typo in Section 4.4.2 has been corrected.
>
> Thank you again for your helpful comments!

---

> > ### Comment · Reviewer_Fo2q · 2024-09-17
> > **Thank you for the response**
> >
> > Thank you for the response;
> > I see that my concerns have been partially addressed in the current version.
> > As the authors say that they will modify the Section 1 contributions and Section 3.2 MQAT overview, and the clarity of the contributions (I think they are two folds) is critical, I would like to wait to see the revised manuscript.  And if that is satisfactory, I don't see any further issue.

---

> > > ### Author Response · Authors · 2024-09-19
> > > **Thank you for your quick response!**
> > >
> > > Thank you for your quick response. As promised, we have revised the manuscript, updating the contributions in Section 1, and significantly enhancing the clarity of the methodology in the entire methods section. We removed section 3.2 as we felt it did not contribute well to emphasizing the original contributions. Instead we reworked 3.1 to better define modularity and replaced all claims of an "optimal quantization flow" to a "recommended quantization flow". However, the heuristic proof of an optimal quantization flow for one case remains in the experiments.

---

> > > > ### Comment · Reviewer_Fo2q · 2024-09-20
> > > > **Concerns have been addressed**
> > > >
> > > > Thank you for the update; it seems the clarity has been significantly improved.
> > > > I don't have any further concern.

---

### Review · Reviewer_ed23 · 2024-09-10

**Summary Of Contributions:**

The paper proposes the Modular Quantization-Aware Training (MQAT) framework where given a modular neural network with different components, quantization is applied to each component in an ordering that will ideally increase the overall performance.

The components are mainly ordered by number of parameters (smallest to largest), with the exception that the module that quantization (to 2 bits) increases the performance the most being placed first (if such a module exists).  This allows for modules with greater number of parameters to be quantized later, as the larger number of parameters will allow for more flexibility to accommodate the quantization of other components. To  determine the quantization level (number of bits to use) for each component, an ILP solver is minimize an overall importance metric (combination of sensitivity and difference between quantized and original weights) across the components.

The proposed training strategy is applied on 6D object pose estimation (evaluated on the LINEMOD, O-LINEMOD, SwissCube datasets).  Experiments show that MQAT results in better performance at higher compression factor compared to two baseline methods (uniform and layer-wise quantization).

The main contribution of the work is the MQAT algorithm (summarized above) and the set of experiments applying it on 6D pose estimation.

**Audience:**

Yes

**Broader Impact Concerns:**

No Broader Impact Statement is provided.

**Claims And Evidence:**

Yes

**Requested Changes:**

While the paper shows the proposed approach can effectively quantize an modular network for 6D pose estimation, some improvements to the paper can be made.
1. Update related work on 6D object pose estimation and quantization to be more current
    - In the related work section (section 2.3), it should be good to discuss CA-SpaceNet [Wang et al. 2022] more and indicate that it is one of the first work to apply quantization to 6D object pose estimation.
2. Provide more details on training and evaluation
    - Splits used for computing accuracy during training
    - Statistics of splits used for training and evaluation
    - Some discussion about the training time (e.g. GPU hours) and what is the overhead of running MQAT
3. Improve discussion of the metrics used
    - Provide a bit more background on ADD / ADI (how is this metric related to ADD-S - is ADD-S similar?).  A brief explanation of the two metric ADD and ADI and intuitively why there are these two metrics (e.g. what acse does ADI handle), and why they are more appropriate than more recent metrics would be useful.
   - Question: As the training algorithm tries to optimize the accuracy, is it overfitting to the metric?
4. Either provide experimental results for updated 6D object pose benchmarks and metrics or provide a description of why those benchmarks are not needed and why the current evaluation on LINEMOD / SwissCube is sufficient.
5. Improve discussion of whether the proposed technique can be applied to different scenarios beyond 6D object pose estimation.  There are some interesting results on object detection in Appendix A.2.  It would be good to have a discussion of it in Section 4 or 5 (not just a sentence in the introduction).
6. Improve citation style and references
    - Currently, the text is just hard to read whenever a reference appears.
        - For example: "such as automated manufacturing Pérez et al. (2016)"
        - Please use proper parenthetical citation (e.g. `\citep`) in such cases so that the text will read "such as automated manufacturing (Pérez et al. 2016)"
    - Also, some references are missing publication venues (e.g. Hu et al. 2019, Kehl et al 2017, and many others), missing author full first name (and thus inconsistent with other entries - see Kehl et al. 2017, Li et al. 2019a, etc), and lacking proper capitalization.
7. Minor: Page 8 "table 1" => "Table 1"

**Strengths And Weaknesses:**

Strengths
- This work appears to be the first work that considers how to order components for quantization
- Experiments indicate the proposed approach is effective
- The paper is mostly well written and easy to follow

Weaknesses
- Related work on 6DOF object pose estimation seems a bit dated (summarizing works only up to 2022)
- Similarly related work on QAT also seems a bit dated (up to 2022 only).  I'm not that familiar with this area but perhaps works like [1] should be discussed.
- Evaluation metrics and datasets for 6D object pose estimation also does not seem to be what is commonly used in recent works.  I'm also not that familiar with the current 6D pose estimation work, but many recent works [2,3] seem to evaluate on the BOP dataset [4].
- Lack of discussion on the time overhead it takes to train with the modular quantization optimization, and the scalability of the approach when the number of modules grows.
- There are some details that are unclear:
  - Algorithm 1:
    - What does `index(SORT_SIZE(B))` do and return?  From examining the algorithm it is likely the indices of the modules sorted by number of parameters (ascending), but it was not that clear.
  - Training and evaluation details
    - What split is used to evaluate the accuracy during training?  The training set, the validation set?
    - How large was the evaluation dataset?
- Some inaccuracies
  - Page 7 states that ADD-0.1 and ADD-0.5 are used.  But the results (Figure 4-6, Table 1,3-6) are for ADI

References

[1] GWQ: Group-Wise Quantization Framework for Neural Networks [Yang et al. ACML 2023]

[2] MRC-Net: 6-DoF Pose Estimation with MultiScale Residual Correlation [Li et al. CVPR 2024]

[3] GigaPose: Fast and Robust Novel Object Pose Estimation via One Correspondence [Nguyen et al. CVPR 2024]

[4] BOP Challenge (https://bop.felk.cvut.cz/home/)

---

> ### Author Response · Authors · 2024-09-15
>
> We sincerely thank the reviewer for their valuable feedback and insightful comments. Below we address each point and highlight the related changes in the revised manuscript.
>
> >**1: Related Work on 6DOF Object Pose Estimation and Quantization**
>
> We appreciate your suggestion to update the related work. In the revised version, we have added some new works on both quantization and object pose estimation, along with the ones you suggested. Moreover, will include a discussion of CA-SpaceNet [Wang et al. 2022], which was one of the first works to apply quantization to 6D object pose estimation. We are currently revising the entire section and will provide an updated version soon. However, it is important to note that our proposed MQAT framework is generic for modular neural networks and can be applied to a wide range of architectures. The modular nature of our method ensures that it works across different architectures, allowing flexibility in adapting to various state-of-the-art models as demonstrated in the provided experiments.
>
> Moreover, we have added [1] in related work, but it does not support mixed precision quantization and does not dynamically assign different bit-widths to different layers or modules. Additionally, the code is not publicly available, making it unfortunately impossible for us to provide results for this method within the given time frame.
>
> [1] GWQ: Group-Wise Quantization Framework for Neural Networks [Yang et al. ACML 2023]
>
>
> >**2. Training and Evaluation Details**
>
> The training and testing splits are consistent with previous work, as outlined in Section 4.1. We would like to clarify that the baseline models were trained and fine-tuned under similar settings, ensuring fair comparisons across all methods. Training times for both MQAT and the baseline models were originally provided in the Appendix; however, we have now included this information in the main paper in Section 4.2.
>
> >**3. Improve discussion of the metrics used ADD and ADI Metrics**
>
> The ADI metric was proposed for image sets with indistinguishable poses, e.g., symmetric objects *[1]. ADI is used here so as to compare fairly with previous works [2]. It seems the literature is converging to the term ADD-S which is mathematically identical to ADI.
>
> [1] Tomas Hodan, Jirı Matas, and Stepan Obdrzalek. On evaluation of 6d object pose estimation.ECCV 2016.\
> [2] Wide-Depth-Range 6D Object Pose Estimation in Space, CVPR 2021.
>
> >**As the training algorithm tries to optimize the accuracy, is it overfitting to the metric?**\
> No, the metric and the loss function are independent functions.
>
> >**4. Use of BOP Dataset and Benchmarks**\
>
> While it is true that recent works often evaluate on the BOP benchmark, we would like to clarify that both LINEMOD and O-LINEMOD, which we have used in our experiments, are in fact part of the BOP dataset [1]. To improve the clarity we will adjust all uses of O-LINEMOD in our paper to LM-O to harmonize with other papers. We selected LM-O for our comparisons as it is the most commonly reported BOP dataset, as also reflected in Table 2(a) of the MRC-Net paper. SwissCube was selected to ensure a fair comparison with [2], one of the pioneering works in applying quantization to 6D object pose estimation.
>
> [1] BOP: Benchmark for 6D Object Pose Estimation. ECCV, 2018.\
> [2] CA-SpaceNet: Counterfactual Analysis for 6D Pose Estimation in Space. IROS, 2022.
>
> >**5. Generalization Beyond 6D Object Pose Estimation**
>
> Thank you for observing our results on object detection in the appendix. Unfortunately, it is not possible to add this to the main body of the paper without removing other results due to the 12 page limitation. Do you think the object detection results should be prioritized over other results such as GDR-Net? Or perhaps simply state that object detection results exist in the appendix?
>
> >**6. Improve citation style and references**
>
> We have revised the paper to maintain a consistent citation style, using proper parenthetical citations (e.g., \citep). Additionally, we ensured that all references are complete and properly formatted, with correct publication venues, full author names, and appropriate capitalization.
>
> Thank you once again for your thorough feedback.

---

> > ### Author Response · Authors · 2024-09-19
> > **Revised Manuscript**
> >
> > Per your feedback we have thoroughly updated the related work section to incorporate more recent advancements in both 6D object pose estimation and quantization-aware training (QAT). In particular, we now discuss CA-SpaceNet [Wang et al. 2022], one of the first works to apply quantization to 6D object pose estimation.This revision ensures that the related work reflects the latest developments in the field, providing a more comprehensive overview of current research. Additionally, we recognize that many recent approaches utilize multi-stage frameworks, which, although more accurate, significantly increase memory usage due to their reliance on multiple networks. For hardware-constrained scenarios, such frameworks are often impractical to deploy, further emphasizing the need for efficient quantization methods like MQAT to ensure both performance and resource efficiency.

---

> > > ### Comment · Reviewer_ed23 · 2024-09-21
> > > **Thank you for the revisions**
> > >
> > > Thank you for the response and the revisions.   The revisions has addressed most of my questions and concerns, and I do not have any other significant questions or concerns.
> > >
> > > Some minor comments:
> > > - Section 3.1: "Architectures for other vision tasks such as object detection or image segmentation contain similarly identifiable modules. Therefore, we also demonstrate the effectiveness of our proposed MQAT method for object
> > > detection, as detailed in the Appendix." => I would recommend the first sentence to be reworded to focus less on "vision tasks" (maybe just omit the word "vision" and just have "tasks"), as it is common to have modules even for non-vision tasks.
> > > - Citation:
> > >    - In some cases, in text citation style (\citet) should be used: "as (Wang et al., 2022)." => "as Wang et al. (2022)", "in (Peng et al., 2019)." =>  "in Peng et al. (2019)."
> > > - Regarding the BOP challenge, it also seems to use different metrics (average recall for over varying thresholds for three different distance metrics).  Are those metrics more standard now for 6D pose estimation or is that just for the BOP challenge?
> > >
> > > The above are just some minor comments / questions.  I do not have any further concerns regarding this work.

---

> > > > ### Author Response · Authors · 2024-09-24
> > > >
> > > > Thank you for the feedback. We have revised the first sentence to focus on "tasks" instead of "vision tasks". Additionally, we have made the necessary adjustments to the citation style, using the in-text format for references like "as Wang et al. (2022)" and "in Peng et al. (2019)."
> > > >
> > > > Yes, we acknowledge that the average recall metric has appeared in the latest BOP challenge alongside the prevalent ADD metric, but it certainly is not as well-established standard as ADD for 6D pose estimation. The papers we reviewed in our work reported ADD/ADD-S and the current state-of-the-art, FoundationPose, also primarily report results using the ADD/ADD-S metric and thus we feel it is the most appropriate choice here.

---

### Decision · Action_Editor_Ki5Y · 2024-10-14

**Recommendation:** Accept as is

**Comment:**

The paper describes an approach to applying mixed precision quantization to various components of a modular neural network architecture for 6D object pose estimation.

The reviewers highlighted the experimental evaluation of the approach, which, despite its simplicity, outperforms carefully selected baselines. Most concerns raised by the reviewers were about the clarity of the paper, but all reviewers acknowledged that the updated version of the paper resolved these issues.

**Audience:**

Yes, I think 6D object pose estimation could be interesting to the TMLR audience. Quantization is a important project especially when deployed in practice.

**Claims And Evidence:**

Claims are well supported by evidence.